# VoxDialogue: Can Spoken Dialogue Systems Understand Information Beyond Words?

**Xize Cheng**[1*]  **Ruofan Hu**[1*]  **Xiaoda Yang**[1]  **Jingyu Lu**[1]  **Dongjie Fu**[1]  **Boyang Zhang**[1]
**Zehan Wang**[1]  **Shengpeng Ji**[1]  **Rongjie Huang**[1]  **Tao Jin**[1]  **Zhou Zhao**[1†]
Zhejiang University[1]   chengxize@zju.edu.cn
Code & Data: `https://voxdialogue.github.io/`

## Abstract

With the rapid advancement of large models, voice assistants are gradually acquiring the ability to engage in open-ended daily conversations with humans. However, current spoken dialogue systems often overlook multi-modal information in audio beyond text, such as speech rate, volume, emphasis, and background sounds. Relying solely on automatic speech recognition (ASR) can lead to the loss of valuable auditory cues, thereby weakening the system's ability to generate contextually appropriate responses. To address this limitation, we propose **VoxDialogue**, a comprehensive benchmark for evaluating the ability of spoken dialogue systems to understand multi-modal information beyond text. Specifically, we have identified 12 attributes highly correlated with acoustic information beyond words and have meticulously designed corresponding spoken dialogue test sets for each attribute, encompassing a total of 4.5K multi-turn spoken dialogue samples. Finally, we evaluated several existing spoken dialogue models, analyzing their performance on the 12 attribute subsets of VoxDialogue. Experiments have shown that in spoken dialogue scenarios, many acoustic cues cannot be conveyed through textual information and must be directly interpreted from the audio input. In contrast, while direct spoken dialogue systems excel at processing acoustic signals, they still face limitations in handling complex dialogue tasks due to their restricted context understanding capabilities.

## 1 Introduction

Voice assistants (Ji et al., 2024a) have rapidly evolved into a focal point of both academic research and industry innovation, aiming to facilitate daily conversations (Li et al., 2017; Lee et al., 2023) and task-oriented dialogues (Budzianowski et al., 2018; Si et al., 2024) with humans. Early iterations relied heavily on automatic speech recognition (ASR) (Cheng et al., 2023b;c; Fu et al., 2024; Lei et al., 2024; Huang et al., 2023), combined with dialogue understanding and state management, to support basic, predefined tasks. However, these systems (Hoy, 2018) were constrained by their limited scope and inability to handle open-ended interactions. The advent of large language models (LLMs) (Touvron et al., 2023) with enhanced understanding and reasoning capabilities has revolutionized voice assistants, enabling them to engage in more dynamic and unrestricted dialogues with users (OpenAI, 2024b). This marks a significant departure from their earlier, more constrained functionalities, opening up new possibilities for human-computer interaction.

Yet, despite these advancements, current spoken dialogue systems (Zhang et al., 2023; Xie & Wu, 2024; Fang et al., 2024; Cheng et al., 2025) often overlook the rich multimodal information embedded in audio beyond mere spoken words—such as intonation, volume, rhythm, and background sounds. Relying solely on ASR leads to the omission of valuable auditory cues, diminishing the system's ability to generate contextually appropriate responses. For example, a system might fail to adjust its language to match a user's emotional state or regional accent, such as responding with "Yes, madam" to a female voice or adopting british colloquialisms when detecting a British accent.

---

[*]Equal Contribution.
[†]Corresponding author.

Table 1: **Comparison of spoken language and audio comprehension benchmarks in terms of data types and evaluation dimensions.** **SL.** refers to Spoken Language, while **Dlg.** indicates whether the benchmark evaluates on dialogue tasks. **Aud.** represents audio comprehension, and **Mus.** refers to music comprehension. **Speaker Info** includes attributes such as age (Age), gender (Gen), accent (Acc), and language (Lan). **Paralinguistic Info** covers aspects like emotion (Emo), volume (Vol), speech rate (Spd), speech fidelity (Fid), stress (Str), and non-verbal expressions (NVE). [†]Although LeBenchmark includes a small amount of conversational data (29 hours out of 2933 hours), it does not evaluate on the dialogue tasks. [‡]Please note that although AirBench can assess spoken language comprehension, its evaluation of conversational ability (AirBench-Chat) is based on text-based interactions and does not address spoken dialogue capabilities.

| Benchmarks | Types | | Evaluation Dimensions | | | |
| --- | --- | --- | --- | --- | --- | --- |
| | SL. | Dlg. | Aud. | Mus. | Speaker Info | Paralinguistic Info |
| SUPERB (Yang et al., 2021) | ✓ | ✗ | ✗ | ✗ | ✗ | ✓ (Emo) |
| SLUE (Shon et al., 2022) | ✓ | ✗ | ✗ | ✗ | ✗ | ✗ |
| LeBenchmark (Evain et al., 2021) | ✓ | ✗[†] | ✗ | ✗ | ✗ | ✓ (Emo) |
| AF-Dialogue (Kong et al., 2024) | ✗ | ✓ | ✓ | ✓ | ✗ | ✗ |
| AirBench (Yang et al., 2024a) | ✗[‡] | ✓ | ✓ | ✓ | ✓ (Age,Gen) | ✓ (Emo) |
| SpokenWOZ (Si et al., 2024) | ✓ | ✓ | ✗ | ✗ | ✗ | ✗ |
| SD-EVAL (Ao et al., 2024) | ✓ | ✓ | ✓ | ✗ | ✓ (Age,Gen,Acc) | ✓ (Emo) |
| VoxDialogue (ours) | ✓ | ✓ | ✓ | ✓ | ✓ (Age,Gen,Acc,Lan) | ✓ (Emo,Vol,Spd,Fid,Str,NVE) |

To address these limitations, recent research has shifted towards developing multimodal audio-language models that enhance system comprehension of audio inputs. Emotion2Vec (Ma et al., 2023), trained on vast emotional speech data, stands as the first high-quality pre-trained model for emotion recognition. Qwen-Audio 1/2 (Chu et al., 2023; 2024) have been trained on extensive datasets encompassing over 30 audio-related tasks, enabling them to understand various audio types—including speech, audio events, and music. Pushing the envelope further, FunAudioLLM (SpeechTeam, 2024) offers full-scene recognition capabilities, detecting non-verbal sounds like laughter and breathing within speech.

As large-scale audio-language models continue to evolve rapidly, the scientific community has increasingly recognized the urgent need for a comprehensive benchmark to effectively evaluate spoken dialogue systems. While some progress has been made, existing benchmarks often exhibit notable shortcomings. For instance, SUPERB (Yang et al., 2021) is the first benchmark specifically designed for spoken language, but it primarily focuses on coarse-grained semantic understanding tasks, overlooking the importance of various acoustic features. Other benchmarks, such as AirBench (Yang et al., 2024a) and Audio-Flamingo (Kong et al., 2024), delve deeply into audio understanding, but their dialogue content is limited to the textual modality, making them unsuitable for evaluating spoken dialogue tasks. SpokenWOZ (Si et al., 2024), though valuable for its real human-computer interaction data, is restricted to task-driven dialogues and lacks detailed fine-grained labels. To address more specific attributes of spoken dialogue, SD-EVAL (Ao et al., 2024) shifts the focus to characteristics like gender, age, accent, and emotion, yet its effectiveness is limited by the use of speech utterances that are not derived from dialogue scenarios.

To better benchmark spoken dialogue systems, we analyzed non-textual multimodal acoustic information that may affect dialogue responses, which can be categorized into three main types: speaker information (*age, gender, accent, language*), paralinguistic information (*emotion, volume, speed, fidelity, stress, and various non-verbal expressions*), and background sounds (*audio and music*). In real-world dialogue scenarios, it is crucial to capture not only the semantic content of the speech but also these acoustic cues to generate more appropriate responses. For example, determining the speaker's age from their vocal tone can help select a suitable form of address. We designed a tailored spoken dialogue synthesis pipeline for each attribute to ensure that the synthesized dialogue data aligns accurately with the corresponding attribute. Leveraging the strong inference capabilities of large language models (LLMs) and high-fidelity text-to-speech (TTS) synthesis (Ji et al., 2024b; Du et al., 2024), we constructed the VoxDialogue benchmark, comprising 12 dialogue scenarios specifically tailored to different acoustic attributes. As shown in Figure 1, to the best of our knowledge, this is the most comprehensive work focusing on acoustic information in spoken dialogue

benchmarks. Based on VoxDialogue, we evaluated several existing spoken dialogue systems, comparing the performance of ASR-based dialogue systems and direct dialogue systems across various acoustic-related tasks. The results demonstrate that ASR-based methods are limited in their ability to understand the diverse acoustic attributes present in spoken dialogues, highlighting the importance of developing large-scale audio-language models. At the same time, existing direct dialogue systems (such as Qwen2-Audio) still exhibit limitations in long-context reasoning, indicating the need for further improvement in their contextual understanding capabilities. All our code and data will be open-sourced. Our main contributions are:

- We present the first benchmark for evaluating the ability of spoken dialogue systems to understand acoustic information beyond speech content, VoxDialogue, which integrates 12 acoustic dimensions, including speaker attributes (*age, gender, accent, language*), paralinguistic features (*emotion, volume, speed, fidelity, stress, non-verbal expressions*), and environmental information (*audio, music*).
- We were the first to develop distinct spoken dialogue data synthesis methods tailored for different acoustic attributes. This approach enables large-scale synthesis of spoken dialogue data, supporting extensive training for spoken dialogue models and endowing them with more comprehensive acoustic understanding capabilities.
- We conducted a systematic evaluation of existing spoken dialogue systems, comparing their performance in terms of understanding acoustic information, supplemented by a qualitative analysis using a GPT-based metric. Specifically, inspired by the MOS (Mean Opinion Score) evaluation mechanism, we provided GPT with descriptive criteria corresponding to different scores, enabling the evaluation model to more accurately assess each response in terms of both acoustic attributes and content quality.

## 2 RELATED WORKS

### 2.1 SPOKEN DIALOG SYSTEM

With the development of large-scale language models, increasingly powerful spoken dialogue models have emerged, utilizing extensive training corpora for single tasks with LLM-based instructions to achieve comprehensive audio understanding capabilities. SpeechGPT (Zhang et al., 2023) integrates discrete speech units into large language models (LLMs), making it a speech-centric model. Qwen-Audio 1/2 (Chu et al., 2023; 2024) established the first large-scale, comprehensive audio model for over 30 audio-related tasks. Similarly, Salmonn (Tang et al., 2023) addresses task complexity in audio models by introducing more intricate story generation tasks. Additionally, some directly use dialogue databases for training. StyleTalk (Lin et al., 2024b) focused on emotional dialogue tasks and introduced the first spoken dialogue model capable of generating responses with varying emotional tones. Recent studies (Cheng et al., 2023c;b;a; Lei et al., 2024; Fu et al., 2024; Yang et al., 2024b) in conversational AI even have begun examining how visual data integration can enhance the contextual awareness of spoken dialogue systems.

However, existing spoken dialogue models (Xie & Wu, 2024; Fang et al., 2024) primarily focus on understanding speech content and audio information, with only a few works specifically addressing detailed acoustic attributes within the speech. This oversight results in the loss of crucial information in spoken dialogue, which, as our experiments show, can significantly undermine the quality and effectiveness of response generation in daily dialogue.

### 2.2 SPOKEN LANGUAGE BENCHMARK

With the rapid development of large-scale audio models (Chu et al., 2024; SpeechTeam, 2024), the scientific community has increasingly recognized the need for a comprehensive benchmark to evaluate spoken dialogue systems. While some progress has been made, many existing benchmarks still fall short. For instance, SUPERB (Yang et al., 2021) was the first benchmark specifically designed for spoken language, but it primarily focuses on coarse-grained understanding tasks. Audio-Flamingo (Kong et al., 2024) explores audio comprehension but is limited to text-based dialogue data, lacking a focus on evaluating spoken dialogue. AirBench (Yang et al., 2024a) was the first to propose using GPT-4 (OpenAI, 2024b) to measure spoken dialogue system performance, but its

Table 2: **Dialogue example of VoxDialog.** The **bold black text** highlights the acoustic and background information unique to spoken dialogue, while the blue text emphasizes responses that depend on these information.

| Attributes | Example |
|---|---|
| *I. Speaker Information* | |
| Gender | A: Excuse me, I'm looking for a reliable smartphone that suits my needs. **(Man Voice)** 
 B: Certainly, Sir! ... excellent performance and a long-lasting battery, perfect for business tasks. |
| Age | A: What dishes do you recommend? **(Young Voice)** 
 B: Sure! You might enjoy our specialty beef burger, it's fantastic with fresh juice. |
| Accent | A: Could you help me with my schedule? **(British Accent)** 
 B: Of course, I'll assist you in organizing your timetable. |
| Language | A: 今天天气怎么样? **(Speaking in Chinese)** 
 B: 今天天气晴朗，气温适中，适合外出活动。 |
| *II. Paralinguistic Information* | |
| Emotion | A: Can you recommend a good movie? **(Sad Tone)** 
 B: I get it. Maybe Soul—it's emotional and might bring you some comfort. |
| Volume | A: Can you tell me how to solve this problem? **(Loud Volume)** 
 B: Could you speak more quietly? People around are studying. |
| Speed | A: I'd like to get the burger, a soda, and also the pasta, please. **(Speaking Quickly)** 
 B: Sorry, could you say that more slowly? I didn't catch everything you said. |
| Fidelity | A: Can you tell me the time of the meeting? **(Poor Audio Quality)** 
 B: I'm having trouble hearing you. Could you improve the audio quality? |
| Stress | A: I really don't like making sushi. **(Emphasis on "making sushi")** 
 B: Oh, I see. If you don't like making sushi, what other types of cooking do you enjoy? |
| Non-verbal Expressions | A: Could you help me move this box? **(With a sigh before speaking)** 
 B: Are you feeling okay? It seems like you're really tired. I can take care of it for you. |
| *III. Environmental Information* | |
| Audio Events | What was that sound just now? **(Background sound: airplane engine sound, explosion sound)** 
 That was a loud explosion. It sounded like the plane exploded. Hope no one was hurt. |
| Music | A: Hey, what instrument is this song played on? **(Music: Piano Song, Sad Song)** 
 B: It should be the piano, it sounds so sad. |

evaluation set remains constrained to QA interactions. SpokenWOZ (Si et al., 2024) is a large-scale task-oriented dataset that offers real human interaction data, making it valuable for evaluating task-driven dialogue systems. SD-Eval (Ao et al., 2024), which emphasizes acoustic attributes such as gender, age, accent, and emotion, uses raw audio from confessional-style corpora, making it less suitable for conversational scenarios.

However, due to the challenges associated with collecting spoken dialogue data in specific scenarios, existing benchmarks are unable to effectively evaluate whether spoken dialogue systems can understand various information beyond words. To address this limitation, we developed VoxDialogue, a benchmark built using synthetic data that focuses on 12 acoustic dimensions that can significantly influence dialogue content. These dimensions include speaker information (age, gender, accent, language), paralinguistic information (emotion, volume, speed, fidelity, stress, non-verbal expressions), and environmental information (audio events, music). Ultimately, VoxDialogue enables a comprehensive evaluation of the ability of current spoken dialogue systems to process and interpret such detailed acoustic information.

# 3 VoxDialogue

## 3.1 Overview

Spoken dialogue systems are typically used in daily dialogues (Lin et al., 2024a). As shown in Table 2, we evaluate the performance of spoken dialogue systems across these three categories in daily dialogue scenarios. Beyond understanding the speech content, spoken dialogue systems must also

generate the most appropriate responses by considering the speaker's emotions, gender, and other acoustic-related information. Therefore, unlike traditional text-based dialogue benchmarks (Li et al., 2017), we systematically analyze the acoustic characteristics that may influence response content and have developed a tailored evaluation set specifically for spoken dialogue systems. The evaluation set for daily dialogue is divided into the following categories: **I. Speaker Information.** (1) *Age*: Responses should be tailored to the speaker's age, adjusting salutations (e.g., Mrs./Miss) or suggesting content appropriate for their age group. (2) *Gender*: Responses should be gender-specific, modifying salutations (e.g., Mr./Mrs.) or offering preferences based on gender. (3) *Accent*: Responses should account for the speaker's accent, selecting vocabulary that aligns with their speech (e.g., British people may be more accustomed to using 'timetable' instead of 'schedule'). (4) *Language*: Responses should be adapted to the speaker's language, choosing the most appropriate language for the response. **II. Acoustic Information.** (5) *Emotion*: Responses should detect the speaker's emotional state and provide a suitable reply (e.g., suggesting comforting music when sensing distress). (6) *Volume*: Responses should consider the speaker's volume, asking them to lower or raise their voice (e.g., requesting quieter speech in quiet environments). (7) *Speed*: Responses should adjust to the speaker's speech rate, asking them to slow down or clarify if speaking too quickly for comprehension. (8) *Fidelity*: Responses should detect poor audio quality and ask the speaker to repeat or improve the clarity of their speech for better understanding. (9) *Stress*: Responses should recognize emphasis on specific words and tailor replies to focus on the stressed content. (10) *Non-verbal Expressions*: Responses should account for non-verbal cues such as sighs, detecting emotions like tiredness or frustration, and offering assistance accordingly. **III. Background Sound.** (11) *Audio Event*: Responses should recognize relevant audio events and adapt accordingly. (12) *Music*: Responses should adjust to the type and mood of the background music.

## 3.2 Spoken Dialogue Generation

**Stage1: Dialogue Script Synthesis.** Building on the methodology of previous studies (Lin et al., 2024a; Cheng et al., 2025), we employed large language models with advanced reasoning capabilities to synthesize spoken conversation scripts tailored to diverse scenarios and acoustic conditions. Specifically, we utilized GPT-4o (OpenAI, 2024a) to pre-generate several rounds of historical conversations, followed by the generation of contextually appropriate responses under various controlled acoustic conditions. This approach ensures that the synthesized dialogue scripts capture a wide range of acoustic features, thereby enhancing their robustness and diversity.

**Stage2: Spoken Dialogue Generation.** We carefully tailored the most appropriate speech synthesis method for each attribute during the generation process We designed a tailored spoken dialogue synthesis pipeline for each attribute to ensure that the synthesized dialogue data aligns accurately with the corresponding attribute: **(1)** *Gender*, *Speed* and *Emotion*. We use COSYVOICE-300M-INSTRUCT[1] to achieve condition speech generation based on gender and emotion by adjusting style instructions. **(2)** *Stress*, *Language*, and *Non-verbal Expressions*. We achieved control over these aspects by adjusting the text content in the COSYVOICE-300M-INSTRUCT (*Stress*, *Non-verbal Expressions*) and COSYVOICE-300M-SFT[2] (*Language*), adding $< stress >< /stress >$, $[laughter]$, or changing the language of the text. **(3)** *Volume*, *Fidelity*, *Audio Events*, and *Music*. We used COSYVOICE-300M-SFT to generate the basic speech, then applied post-processing techniques to fine-tune these specific attributes. The details of post-processing are shown in Stage 4. **(4)** *Age*. We randomly selected 1,000 speaker samples of different ages from Hechmi et al. (2021) and Tawara et al. (2021) as reference timbres and used COSYVOICE-300M[3] for zero-shot TTS synthesis. **(5)** *Accent*. We used the industrial-grade TTS tool (edge-TTS[4]), which offers over 318 timbre references spanning various regions, languages, and genders to achieve precise accent generation.

**Stage3: Automatic Verification for Spoken Dialogue.** To ensure the quality of the synthesized spoken dialogue data, we first employed a pre-trained model to automatically filter out unqualified samples, removing those with generation errors and inconsistent timbre. Specifically, we used the Whisper model (Radford et al., 2023) to filter out all sentences with a word error rate (WER) greater

---

[1]https://huggingface.co/FunAudioLLM/CosyVoice-300M-Instruct
[2]https://huggingface.co/FunAudioLLM/CosyVoice-300M-SFT
[3]https://huggingface.co/model-scope/CosyVoice-300M
[4]https://github.com/rany2/edge-tts

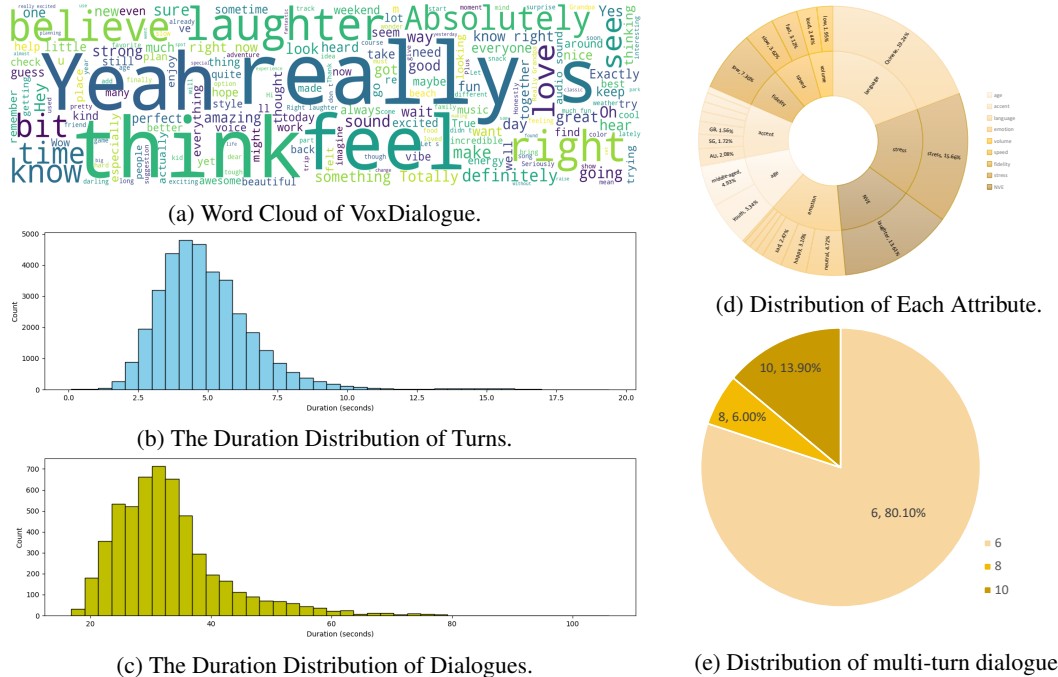

(a) Word Cloud of VoxDialogue.

(b) The Duration Distribution of Turns.

(c) The Duration Distribution of Dialogues.

(d) Distribution of Each Attribute.

(e) Distribution of multi-turn dialogue.

Figure 1: Visualization of static analysis of VoxDialogue.

than 5%, and applied speaker-diarization-3.1 (Plaquet & Bredin, 2023; Bredin, 2023) to eliminate samples with timbre inconsistencies in speeches of the same speaker throughout dialogue sequence.

**Stage4: Post-processing for Specific Acoustic Attributes.** For attributes such as volume, fidelity, audio events, and music, we performed post-processing to ensure that the audio aligns with the required expectations. For *fidelity*, according to the Nyquist-Shannon sampling theorem, the sampling rate must be at least twice the highest frequency of the signal to ensure lossless reconstruction. To capture frequencies up to 4 kHz, the minimum sampling rate should be 8 kHz. Therefore, we downsampled the speech to 4 kHz (to simulate the loss of speech signal and represent 'poor' audio quality, resulting in the loss of some speech information) and then resampled it back to 16 kHz to simulate poor audio fidelity. For *volume*, dialogue turns labeled as *'loud'* were amplified to simulate by increasing the power 8-fold. For dialogue turns labeled as *'low'*, the audio power was reduced to 50% of its original level to simulate poor microphone reception. For *audio events*, a large language model is used to classify events as either temporary or continuous. Temporary audio events, such as a door slamming or a phone ringing, are brief sounds that occur momentarily and are spliced before the first voice segment. In contrast, continuous audio events, like background chatter or street noise, are prolonged and are looped as background sound throughout the conversation. For **music**, we randomly spliced it before the first speech segment or set it to play in a loop as background sound.

**Stage5: Human Verification.** While large language models (LLMs) are effective at following instructions and generating coherent conversation samples, they are primarily trained on text data and lack exposure to human spoken conversations. As a result, the automatically generated data may exhibit unnatural characteristics. To ensure the naturalness and logical consistency of the spoken conversation sample pairs with the audio features, we employ human annotators for additional quality checks.

### 3.3 DATASET STATISTICS

**Distribution of Attribute Categories.** As shown in Figure 1 (d), the distribution of attribute categories in VoxDialogue is balanced, allowing for a comprehensive evaluation of spoken dialogue systems' understanding and dialogue capabilities across various acoustic attributes. In Figure 1 (a), we also present a word cloud of VoxDialogue, where it is evident that the dataset primarily con-

Table 3: Detailed statistics of the corresponding subsets of each attribute in VoxDialogue. Gray fonts indicate that samples of this attribute are included in other subsets. IN (India), CA (Canada), ZA (South Africa), GB (United Kingdom), SG (Singapore), US (United States), and AU (Australia). **Turns** represents the total number of turns in each subset, **Dialog.** indicates the number of dialogues in each subset, **Avg** denotes the average number of turns per dialogue in each subset, and **Dur.** refers to the total duration (in hours) of all dialogues in each subset.

| Attributes | Categories | Turns | Dialog. | Avg | Dur. |
|---|---|---|---|---|---|
| *I. Speaker Information* | | | | | |
| Gender | Male, Female | 2040 | 340 | 6.0 | 3.17 |
| Age | Youth (15-30), Middle-Aged (30-60), Elderly (60+) | 3096 | 447 | 6.9 | 6.05 |
| Accent | IN, CA, ZA, GB, SG, US, AU | 1440 | 240 | 6.0 | 2.20 |
| Language | Chinese, English | 2892 | 482 | 6.0 | 3.51 |
| *II. Paralinguistic Information* | | | | | |
| Emotion | Neutral, Happy, Sad, Angry, Surprised, Fearful, Diagusted | 1980 | 330 | 6.0 | 2.41 |
| Volume | Loud Volume, Low Volume, Normal Volume | 1824 | 304 | 6.0 | 2.08 |
| Speed | High Speed, Low Speed, Normal Speed | 2184 | 364 | 6.0 | 2.93 |
| Fidelity | Low Fidelity, Normal Fidelity | 2196 | 366 | 6.0 | 3.36 |
| Stress | Stress, No Stress | 2354 | 392 | 6.0 | 2.51 |
| NVE | Laughter, No Laughter | 2046 | 341 | 6.0 | 3.68 |
| *III. Environmental Information* | | | | | |
| Audio | The caption of different audio. (e.g., The wind is blowing and rustling occurs.) | 5000 | 500 | 10.0 | 5.25 |
| Music | The aspect list of different music pieces. (e.g., [steeldrum, higher register, amateur recording]) | 3734 | 420 | 8.9 | 5.42 |
| **Overall** | | **30.7K** | **4.5K** | **6.8** | **42.56** |

sists of daily dialogue, featuring a large number of natural spoken words such as "yeah," which are representative of daily spoken interactions. This makes it suitable for assessing the performance of spoken dialogue systems in real-world dialogue scenarios. Additionally, the dataset contains numerous acoustically relevant keywords, such as "heard," "loud," and "sound," further supporting the evaluation of acoustic-related aspects of dialogue understanding.

**Distribution of Dialogue Turns and Duration.** All dialogues in our dataset are multi-turn dialogues. In Figure 1 (e), we show the distribution of dialogue turns, with the majority consisting of 6 turns and a maximum of 10 turns. This allows for a comprehensive evaluation of spoken dialogue systems' ability to understand contexts of varying lengths. In addition, Figures 1 (b) and 1 (c) illustrate the distribution of each turn and the overall dialogue length, respectively, showing that most sentences are approximately 4 seconds long. This implies that the system must understand the context and reason effectively before generating a response.

**Statistics for Subset of Each Attribute.** We present the detailed statistics of each attribute in VoxDialogue in Table 3, covering 35 different categories across 12 attributes. The average number of turns per dialogue exceeds 6, with each attribute containing more than 300 dialogues, ensuring comprehensive reflection of dialogue capabilities.

## 4 BENCHMARK FOR SPOKEN DIALOGUE SYSTEM

### 4.1 TASK DEFINITION

The task of a spoken dialogue system is to generate appropriate responses based on the contextual information from the sequence of human dialogue (e.g., the user's utterance sequence) and the preceding assistant response sequence, where the total number of dialogue turns is denoted by $t$. The goal of the spoken dialogue system is to generate the most suitable response based on the previous

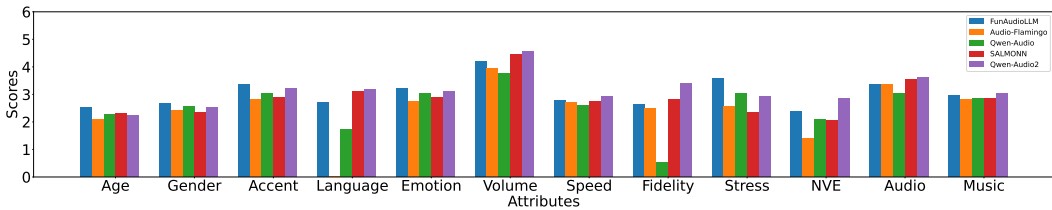

(a) Comparison of BLEU Across Methods and Attributes.

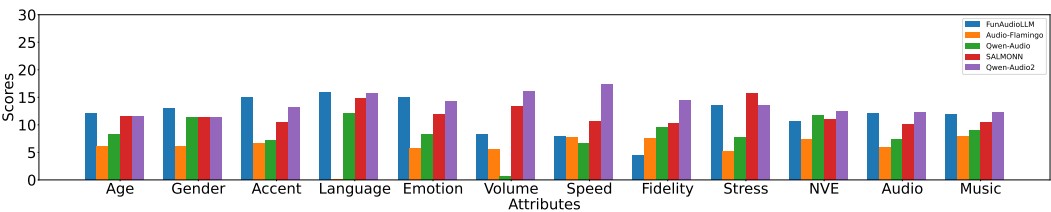

(b) Comparison of ROUGE-L Across Methods and Attributes.

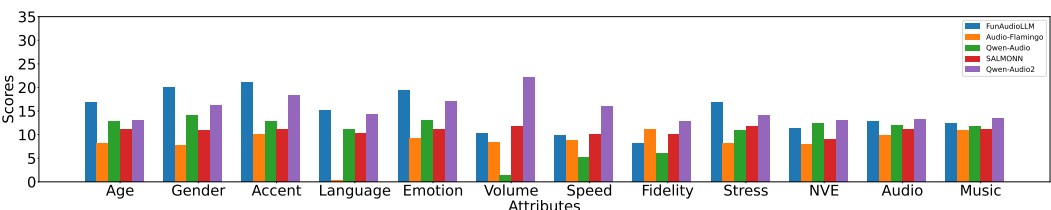

(c) Comparison of METEOR Across Methods and Attributes.

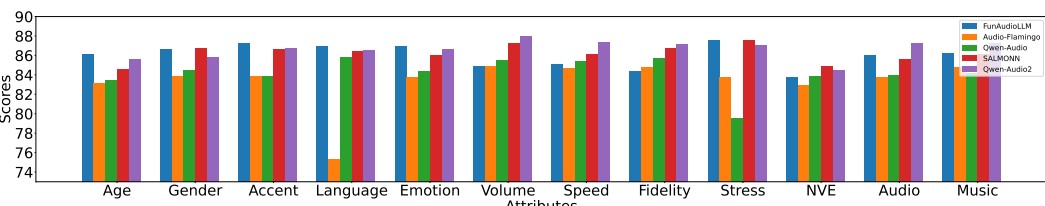

(d) Comparison of BERTScore Across Methods and Attributes.

Figure 2: The comparison of spoken dialogue performance across 12 different attribute-specific test sets on the VoxDialogue dataset.

$t$ utterances and the $t − 1$ historical replies. In our work, we evaluate the performance of the spoken dialogue system by focusing solely on the final utterance of each dialogue.

## 4.2 EVALUATION METRICS

To assess the model's performance, we conducted separate tests on a subset of Voxdialogue. Drawing on previous research (Ao et al., 2024), we utilized both quantitative and qualitative metrics for a comprehensive evaluation. The quantitative evaluation focused on two key aspects: content and style. For content evaluation, we employed widely recognized text generation metrics, including vocabulary-level measures such as BLEU (Papineni et al., 2002), ROUGE-L (Lin, 2004), and ME-TEOR (Banerjee & Lavie, 2005), alongside semantic-level metrics like BERTScore (Zhang et al., 2019). For style evaluation, we calculated the weighted F1 score of speech sentiment.

In addition to these quantitative assessments, we conducted a qualitative analysis using GPT-based metric (Yang et al., 2024a). The meaning of each score is as follows: **1**: Contextually relevant but lacks attribute information. **2**: Partially relevant to the context but feels unnatural, with no attribute information. **3**: Partially relevant to the context, with mention of the attribute. **4**: Contextually

Table 4: GPT-based Metric Comparison of Different Spoken Dialogue Models on VoxDialogue.

| Method | Speaker Info | | | | Paralinguistic Info | | | | | | Env Info | |
|---|---|---|---|---|---|---|---|---|---|---|---|---|
| | Age | Gen | Acc | Lan | Emo | Vol | Spd | Fid | Str | NVE | Aud | Mus |
| *ASR-Based Spoken Dialogue System* | | | | | | | | | | | | |
| FunAudioLLM (SpeechTeam, 2024) | **4.32** | **4.39** | **3.57** | **4.61** | **4.09** | 1.82 | 1.92 | 1.79 | 3.13 | 2.87 | 3.47 | 3.59 |
| *Direct Spoken Dialogue System* | | | | | | | | | | | | |
| Audio-Flamingo (Kong et al., 2024) | 1.00 | 1.00 | 1.04 | 1.72 | 1.00 | 1.20 | 1.14 | 1.26 | 1.34 | 3.06 | 1.37 | 1.11 |
| SALMONN (Tang et al., 2023) | 1.99 | 1.64 | 1.78 | 3.50 | 1.84 | 2.88 | 2.27 | 2.29 | 3.86 | 2.59 | 2.15 | 2.23 |
| Qwen-Audio (Chu et al., 2023) | 1.36 | 1.04 | 1.28 | 1.04 | 1.06 | 1.48 | 1.08 | 1.32 | 2.49 | 2.65 | 1.42 | 1.18 |
| Qwen2-Audio (Chu et al., 2024) | 3.46 | 4.18 | 2.71 | 4.43 | 3.73 | **3.06** | **3.29** | **2.98** | **3.93** | **3.46** | **3.81** | **3.98** |

relevant and natural, mentioning the attribute, but could be improved. **5**: Contextually relevant, smooth, natural, and accurately addresses the attribute. We have included all the evaluated prompt templates in supplementary materials. Please refer to the supplementary materials for more details.

### 4.3 SPOKEN DIALOGUE SYSTEM

In order to build a comprehensive benchmark, we evaluated two main types of spoken dialogue system approaches: (1) **ASR-based dialogue systems** (e.g., FunAudioLLM (Fang et al., 2024)) and (2) **direct spoken dialogue systems**[5] (e.g., Audio-Flamingo (Kong et al., 2024), SALMONN (Tang et al., 2023), Qwen-Audio Instruct (Chu et al., 2023), and Qwen2-Audio Instruct (Chu et al., 2024)). Figure 2 presents a comparative analysis using four metrics across various attributes on the VoxDialogue dataset. Based on the experimental results, we gained the following key insights:

**ASR-based systems excel in context-sensitive tasks.** In attributes that can be inferred through context understanding, ASR-based systems (such as FunAudioLLM) show significant advantages. ASR systems first transcribe speech into text and then process it, allowing them to more effectively capture and analyze the context of a conversation. For example, in attributes like *Emotion* and *Speaker Information(Age, Gender, Accent, Language)*, FunAudioLLM consistently outperforms direct spoken dialogue systems. The results from BLEU, ROUGE-L, METEOR, and BERTScore metrics indicate that FunAudioLLM achieves higher scores, such as in emotion (3.22 BLEU, 14.93 ROUGE-L, 19.31 METEOR, 86.92 BERTScore). This proves that most current direct spoken dialogue systems lack adequate context understanding capabilities and are far weaker than text-based large language models. Additionally, although ASR-based models may have limitations in understanding acoustic information, comparing them provides a valuable performance reference, representing the upper bound performance without the integration of acoustic information.

**Advantages of direct spoken dialogue systems in acoustic attribute processing.** Although ASR-based systems can leverage the strong context understanding capabilities of large language models, they struggle with attributes that heavily rely on sound understanding (such as volume, fidelity, speed, and other paralinguistic information). ASR-based methods face challenges when addressing dialogue tasks related to these attributes. In contrast, direct systems like Qwen2-Audio excel in tasks involving these acoustic properties. The results show that Qwen2-Audio outperforms other systems in these categories. For instance, Qwen2-Audio achieved the highest scores for *volume* (4.56 BLEU, 16.13 ROUGE-L, 22.82 METEOR, and 87.99 BERTScore), demonstrating its ability to handle loud and soft speech variations more effectively. Similarly, *fidelity* is another strong point for direct dialogue systems. Qwen2-Audio's excellent performance in handling varying fidelity levels (3.38 BLEU, 14.36 ROUGE-L, 12.78 METEOR, 85.66 BERTScore) confirms that spoken dialogue tasks, which heavily rely on acoustic information beyond words.

### 4.4 QUALITATIVE COMPARISON

Inspired by Yang et al. (2024a), we also attempted to use GPT-4 (OpenAI, 2024b) for evaluation, focusing on whether the responses exhibit the specific attribute characteristics and whether they

---

[5]All models used in the evaluation are *-chat* version.

provide reasonable replies to the previous context. As shown in Table 4, we present the qualitative testing results of different methods across 12 attributes. Specifically, a score of 3 represents mention of attribute information, 4 represents a reasonable and natural response.

We observed that the conclusions from the qualitative tests largely align with those from the quantitative evaluations. For context-driven attributes (such as speaker information and emotion), ASR-based dialogue models continue to demonstrate the best performance. However, for attributes that are highly dependent on acoustic information (such as speed, fidelity, audio, and music), direct spoken dialogue models like Qwen2-Audio significantly outperform FunAudioLLM, underscoring the importance of developing direct spoken dialogue models.

Additionally, we found that Qwen-Audio often responds with descriptive sentences related to the query, which severely affects its performance. The SALMONN model frequently repeats parts of the query, leading to higher quantitative scores in some attributes (e.g., a BLEUScore of 87.53 for Stress, 0.53 higher than Qwen2-Audio), but its qualitative performance is inferior to Qwen2-Audio (with a GPT-4-based metric score 0.97 lower). This indicates that most current large audio-language models are focused on QA-style interactions, and are not yet well-suited for dialogue-style conversations.

## 5 ETHICAL DISCUSSION

Our dataset incorporates certain attributes that may introduce bias (e.g., gender) as dimensions to evaluate the model's ability to process diverse acoustic information. However, this introduces potential risks of unfairness, such as biased or stereotypical responses. In Appendix C, we outline the fairness challenges faced by spoken dialogue models.

To promote the development of unbiased spoken dialogue systems, we conducted manual filtering of all potentially sensitive data to ensure that the dataset complies with Collins & Clément (2012) and excludes examples that could cause harm due to attribute-related biases. Looking ahead, we are committed to proactively identifying and addressing these challenges, contributing to the creation of fairer and more inclusive spoken dialogue systems. Furthermore, we pledge to continually update this work to advance the development of equitable conversational AI.

## 6 CONCLUSION

In this work, we introduced **VoxDialogue**, a comprehensive benchmark designed to evaluate spoken dialogue systems' ability to understand information beyond words. By identifying 12 critical attributes tied to acoustic cues such as speech rate, volume, emphasis, and background sounds, we constructed a challenging test set of 4.5K multi-turn dialogue samples. Our experiments demonstrated that while ASR-based systems excel at context understanding and textual interpretation, they fail to capture important acoustic signals that are essential for contextually appropriate responses. In contrast, direct spoken dialogue systems outperform ASR-based models in processing acoustic properties, but their limited ability to understand complex dialogue contexts remains a significant shortcoming. The findings highlight the importance of acoustic information in enhancing the performance of spoken dialogue systems and reveal the current limitations in both ASR-based and direct spoken dialogue models.

### REPRODUCIBILITY STATEMENT

All of our data, code, and model weights will be open-sourced.

- Section 3 provides detailed instructions on the construction of VoxDialogue, including a comprehensive list of all relevant open-source resources.

- Section 4.1 outlines the detailed task definitions.

- Section 4.2 elaborates on the evaluation metrics and specific details.

- All of our prompt templates are included in the Supplementary Material.

## ACKNOWLEDGMENTS

This work was supported in part by National Natural Science Foundation of China under Grant No. 62222211 and No.624B2128.

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

Table 5: Detailed Comparison of Spoken Dialogue Systems across Various Metrics

(a) BLEU Scores

| Method | Speaker Info | | | | Paralinguistic Info | | | | | | Background | |
|---|---|---|---|---|---|---|---|---|---|---|---|---|
| | Age | Gen | Acc | Lan | Emo | Vol | Spd | Fid | Str | NVE | Aud | Mus |
| FunAudioLLM | 2.53 | 2.66 | 3.34 | 2.72 | 3.22 | 4.20 | 2.77 | 2.65 | 3.58 | 2.37 | 3.34 | 3.24 |
| Audio-Flamingo | 2.08 | 2.40 | 2.83 | 0.01 | 2.74 | 3.95 | 2.70 | 2.50 | 2.58 | 1.41 | 3.38 | 2.81 |
| Qwen-Audio | 2.26 | 2.56 | 3.05 | 1.74 | 3.01 | 3.78 | 2.61 | 0.54 | 3.02 | 2.85 | 3.60 | 2.87 |
| SALMONN | 2.29 | 2.35 | 2.88 | 3.09 | 2.88 | 4.44 | 2.73 | 2.82 | 2.33 | 2.04 | 3.55 | 2.86 |
| Qwen2-Audio | 2.22 | 2.52 | 3.20 | 3.18 | 3.11 | 4.56 | 2.92 | 3.38 | 2.93 | 2.10 | 2.97 | 2.97 |

(b) ROUGE-L Scores

| Method | Speaker Info | | | | Paralinguistic Info | | | | | | Background | |
|---|---|---|---|---|---|---|---|---|---|---|---|---|
| | Age | Gen | Acc | Lan | Emo | Vol | Spd | Fid | Str | NVE | Aud | Mus |
| FunAudioLLM | 12.15 | 12.95 | 15.07 | 15.88 | 14.93 | 8.28 | 7.97 | 4.47 | 13.49 | 10.67 | 12.01 | 11.97 |
| Audio-Flamingo | 6.12 | 6.15 | 6.62 | 0.03 | 5.78 | 5.48 | 7.67 | 7.57 | 5.12 | 7.41 | 5.91 | 7.88 |
| Qwen-Audio | 8.34 | 9.62 | 7.12 | 12.09 | 8.24 | 0.71 | 6.61 | 14.36 | 7.76 | 12.36 | 7.29 | 9.01 |
| SALMONN | 11.52 | 11.43 | 10.51 | 14.80 | 11.81 | 13.30 | 10.56 | 10.22 | 15.71 | 11.01 | 10.05 | 10.51 |
| Qwen2-Audio | 11.51 | 11.44 | 13.18 | 15.66 | 14.18 | 23.13 | 17.34 | 9.58 | 13.45 | 11.36 | 12.23 | 12.18 |

(c) METEOR Scores

| Method | Speaker Info | | | | Paralinguistic Info | | | | | | Background | |
|---|---|---|---|---|---|---|---|---|---|---|---|---|
| | Age | Gen | Acc | Lan | Emo | Vol | Spd | Fid | Str | NVE | Aud | Mus |
| FunAudioLLM | 16.89 | 20.12 | 21.03 | 15.21 | 19.31 | 10.19 | 9.83 | 8.16 | 16.95 | 10.31 | 12.91 | 12.42 |
| Audio-Flamingo | 8.23 | 7.79 | 10.03 | 0.25 | 9.17 | 8.31 | 8.69 | 11.04 | 8.12 | 7.88 | 9.93 | 11.01 |
| Qwen-Audio | 12.87 | 14.16 | 12.92 | 11.06 | 13.12 | 1.41 | 5.28 | 6.11 | 10.92 | 21.41 | 11.92 | 11.68 |
| SALMONN | 11.02 | 10.81 | 11.21 | 10.35 | 11.13 | 11.78 | 10.14 | 10.17 | 11.84 | 9.03 | 11.18 | 11.08 |
| Qwen2-Audio | 12.96 | 16.15 | 18.24 | 14.37 | 17.05 | 22.11 | 19.08 | 12.78 | 14.01 | 13.11 | 13.21 | 13.39 |

(d) BERTScore

| Method | Speaker Info | | | | Paralinguistic Info | | | | | | Background | |
|---|---|---|---|---|---|---|---|---|---|---|---|---|
| | Age | Gen | Acc | Lan | Emo | Vol | Spd | Fid | Str | NVE | Aud | Mus |
| FunAudioLLM | 86.14 | 86.65 | 87.24 | 86.97 | 86.90 | 84.87 | 85.03 | 84.36 | 87.51 | 83.79 | 86.02 | 86.19 |
| Audio-Flamingo | 83.10 | 83.84 | 83.86 | 75.28 | 83.78 | 84.91 | 84.71 | 84.81 | 83.78 | 82.89 | 83.78 | 84.74 |
| Qwen-Audio | 83.40 | 84.46 | 83.84 | 85.79 | 84.34 | 85.53 | 85.34 | 87.12 | 79.55 | 83.85 | 83.95 | 84.14 |
| SALMONN | 84.60 | 86.75 | 86.65 | 86.44 | 86.05 | 87.27 | 86.06 | 86.74 | 87.53 | 84.92 | 85.63 | 86.06 |
| Qwen2-Audio | 85.59 | 85.80 | 86.70 | 86.50 | 86.65 | 88.00 | 87.30 | 85.66 | 87.08 | 84.51 | 87.22 | 87.19 |

## A    MORE EXPERIMENT RESULTS

### A.1    THE DETAILED PERFORMANCE COMPARISON

For comparison, the detailed performance corresponding to Figure 2 is presented in Table 5.

## B    LIMITATION

Our work heavily relies on synthetic datasets. Although prior research (Liu et al., 2023) has shown that synthetic data can be effectively used for training and evaluation, a domain gap persists between

synthetic and real-world data. This gap may affect the generalization of models trained on synthetic data when applied to real-world dialogue scenarios.

However, since our focus is on understanding acoustic information, synthetic data proves particularly useful in simulating various acoustic cues found in real conversational settings. Additionally, the synthetic dataset offers more diverse and controllable dialogue content, making it sufficient for evaluating whether spoken dialogue systems can understand information beyond text.

To properly assess the performance of dialogue systems in real-world scenarios, it is crucial to use datasets based on authentic conversational environments. We believe that constructing a separate real-world dialogue evaluation benchmark, independent of our work, would be more effective in evaluating spoken dialogue systems' performance in real scenarios than using a single dataset to assess both acoustic information comprehension and real-world dialogue capabilities.

## C  ETHICAL DISCUSSIONS ON SPOKEN DIALOGUE SYSTEMS

### C.1  FAIRNESS CHALLENGES IN SPOKEN CONVERSATION.

Ensuring fairness in spoken dialogue systems involves several challenges, particularly when addressing attributes like gender:

**I. Difficulty in Identifying Gender Bias:** Beyond explicit expressions (e.g., "sir" or "madam"), implicit biases (e.g., career or family-related topics) (Liu et al., 2019) are deeply embedded in existing large language models, making it difficult to guarantee unbiased responses.

**II. Fairness and Attribute Understanding:** While understanding gender attributes can enhance personalization and conversational relevance, over-reliance may reinforce stereotypes. Conversely, completely eliminating gender considerations could limit the model's ability to provide contextually appropriate responses in scenarios where gender information is explicitly relevant. Therefore, an appropriate balance between fairness and attribute understanding should be achieved, ensuring that biases do not cause harm while fostering diversity in responses and improving attribute-specific relevance.

**III. Difficulty in Evaluating Bias:** Current fairness metrics (Su et al., 2023) often fail to capture nuanced and context-dependent biases in spoken dialogue systems, especially in open-ended and multi-turn conversations.

### C.2  MITIGATION STRATEGIES.

To address these challenges, we commit to implementing the following measures:

**I. Manual Filtering:** We conducted manual filtering of all potentially sensitive data to ensure that the dataset complies with Collins & Clément (2012) and excludes examples that could cause harm due to attribute-related biases.

**II. Bias Warnings:** Clear disclaimers will be included in the documentation to highlight potential gender biases and encourage developers to consider fairness during model development.

**III. Continuous Dataset Updates:** We will continuously update and refine the dataset to address fairness issues. Any subsets or evaluation components found to introduce risks of bias will be removed or adjusted as necessary.

