# OpenReview forum: "VoxDialogue: Can Spoken Dialogue Systems Understand Information Beyond Words?"
_ICLR.cc/2025/Conference — ICLR 2025 Poster_

### Official Review · Reviewer_CNoA · 2024-10-24

**Soundness:** 3
**Presentation:** 3
**Contribution:** 3
**Rating:** 6
**Confidence:** 4

**Summary:**

The authors present a dataset called VoxDialogue to benchmark the ability of spoken dialogue systems to leverage acoustic information to adapt their interaction. Using LLMs and TTS, they created 4.5K multi-turn spoken dialogue samples according to 12 factors (e.g. age, language, accent, volume, noise...). The authors then evaluated several existing spoken dialogue systems on this benchmark showing that such systems struggle in these situations given their low BLEU and ROUGE scores.

**Strengths:**

Making available such datasets  with controlled conditions over 12 factors (Gender, Speed, Emotion, Stress, Language, Non-verbal Expressions, Volume, Fidelity, Audio Events Music, Age, and Accent) is a very interesting contribution to the field. The creation process has been performed with care using the latest industrial methods (GPT-4o and GTP4Microsoft Edge's online text-to-speech).

The evaluation of the five systems (Audio-Flamingo, Qwen-Audio, SALMONN, Qwen-Audio2 and FunAudioLLM) is also very interesting, and forms the basis of the comparative analysis.

**Weaknesses:**

The paper does not present technical novelty nor original metrics to evaluate dialogue systems. For instance, the evaluation has been performed using n-gram metrics which fails to handle variations in answers. BERT-Score is  similar in all conditions and error bar is not provided. This makes it difficult to assess the difference between models. The evaluation with GPT4 is correlated to other metrics but with more extreme differences. Human evaluation would definitely be an added value here.

As raised by the authors using TTS and LLM-generated content might be too far from realistic settings. The benchmark might thus be useful for developing systems rather evaluating them. It is true that TTS is useful for training models (Liu et al. 2024 was about lip movements generation) it can be harmful when it is the only data available (Desot et al. 2020, Corpus generation for voice command in smart home and the effect of speech synthesis on End-to-End SLU). But if that informs about the training, it does not support using it for evaluation.

**Questions:**

The generated dialogues I have listened and seen do not seem to have a goal (not task oriented) hence it is difficult to understand how a system should answer for reached a target.

The other question is related to the choice of the human behind. Assessing a gender from voice can be a bad idea simply because the system could be wrong and also because the human behind might not agree being attributed a title related to a gender.

Some of the attributes (age for instance) might be seen as too much intrusiveness. What is the stance of the authors about this?

How many languages have been considered in the datasets?


details

SUPERB is not the only benchmark in 2021, Lebenchmark (Evain et al. 2021), was also there

there are repetitions of sentences between the introduction and section 2

**Details Of Ethics Concerns:**

the fact that the system must use a title Mr/Ms without asking the human behind can be seen as offensive. Not sure it is a good idea that a benchmark contains such behavior.

---

> ### Author Response · Authors · 2024-11-22
> **Response to Reviewer CNoA**
>
> Thank you for your recognition of our work and for acknowledging the value of the 12 acoustic factors in the field. Allow me to clarify some of your questions:
>
> **Q1**: Original Metrics to Evaluate Dialogue Systems
>
> **A1**: To better evaluate the ability of spoken dialogue systems to focus on each attribute, we have made certain improvements to the GPT-based metric. In previous studies[1,2], GPT-based evaluation metrics were often vague and lacked clear standards, resulting in assessments that did not adequately focus on specific acoustic information. Inspired by the way human MOS (Mean Opinion Score) evaluations require descriptive explanations for different score levels, our work introduced detailed scoring guidelines for GPT (as mentioned in the second paragraph of Section 4.2), enabling it to provide more systematic and attribute-specific evaluations.
>
> **Q2**: Human Evaluation
>
> **A2**: To ensure consistent evaluation standards across different researchers, we opted for the improved GPT-based evaluation method rather than human evaluation. Since human evaluators may carry personal preferences, human evaluation often carries a greater risk of inconsistent scoring, which can undermine the reliability of assessments. This decision was made to address potential reproducibility concerns.
>
> To meet your expectations, we are currently conducting human evaluations of the experimental results and will present them before the discussion concludes. If you have any suggestions for further improving our benchmark evaluation system while ensuring standard reproducibility, we would greatly appreciate your insights and look forward to discussing them further.
>
> **Q3**: Discussion on Gendered Titles and Potential Offense
>
> **A3**: Thank you for your in-depth consideration of our dataset. Using gendered titles can indeed raise concerns about potential offense. However, **it is important to note that spoken dialogue is a one-to-many problem, and our provided answers serve only as reference responses, not definitive ones.** Beyond titles like Mr./Ms., the dataset includes many other gender-related information in the responses. As long as the other content aligns with the context, the performance metrics of the model should not be significantly impacted.
>
> It is also worth noting that other widely used datasets, such as SpokenWoz[3], adopt similar approaches by including gendered titles like Mr./Ms. This type of bias has been present in many existing datasets and models, reflecting broader challenges in the field. However, we believe it is essential to separate the discussion of potential bias from the primary goal of evaluating acoustic attributes.
>
> To date, there has been no systematic discussion of potential biases in spoken dialogue datasets, and many biases may remain subtle and difficult to detect. Therefore, it is currently impossible to guarantee that our dataset is entirely free from bias. However, we are committed to systematically reconstructing the dataset to ensure fairness once a comprehensive review of bias phenomena becomes available.
>
> Additionally, as an evaluation dataset, our model does not participate in the training process, which minimizes the risk of propagating biases to new spoken dialogue models. While we recognize that this does not entirely eliminate the issue, it ensures that the potential impact is manageable at present. We remain open to further feedback and discussions on how to enhance fairness in our dataset and beyond.
>
> **Q4:** How many languages are included?
>
> **A4**: Currently, the dataset includes only two languages (Chinese and English), as these are the two primary languages used in most spoken dialogue models. Due to the significant variability in how different models perform with other languages, adding additional languages would require designing separate evaluation metrics for each language to effectively and fairly measure model performance on a per-language basis. This might be better suited as a separate study specifically aimed at evaluating models’ capabilities in handling multiple languages.
>
> **Q5**: Other Issues
>
> **A5**: We have added LeBenchmark in Table 1.
>
> Thank you once again for your insightful discussion of our work. Your feedback has been invaluable, and we hope this response adequately addresses your concerns. If you have any further questions or suggestions, we would be delighted to engage in continued dialogue.
>
> [1] Yang Q, Xu J, Liu W, et al. AIR-Bench: Benchmarking Large Audio-Language Models via Generative Comprehension[J]. ACL2024
>
> [2] Ao J, Wang Y, Tian X, et al. SD-Eval: A Benchmark Dataset for Spoken Dialogue Understanding Beyond Words[J]. NeurIPS2024
>
> [3] Si S, Ma W, Gao H, et al. Spokenwoz: A large-scale speech-text benchmark for spoken task-oriented dialogue agents[J]. NeurIPS2023

---

### Official Review · Reviewer_NcGN · 2024-11-02

**Soundness:** 2
**Presentation:** 3
**Contribution:** 3
**Rating:** 8
**Confidence:** 4

**Summary:**

This paper contributes to the design and creation of a novel dataset for dialogue systems benchmarking. Its design and creation relies on synthetic data and its main attempt is to close the gap on paralinguistic information understanding which its more appropriate to infer directly from acoustics than from the recognized text.

The paper summaries the current gap in SOTA benchmarks and the limitations of the existing data sets and evaluation protocols. This contribution is oriented for enhancing dialogue understanding. The protocol and steps how Voicebox is created is detailed into steps and quantitative and qualitative assessed.

**Strengths:**

This paper presents a solid contribution to standardize, democratize, and accelerate the evaluation of spoken dialogue systems. The summary of the state of the art and status quo is well described and the voice attributes and dialogue rubrics is well covered. Authors also commit to open source their protocol and database, which will help the community to iteratively improve it beyond the current contribution. The mental model proposed on synthetic data is a strong tool to accelerate the capabilities of dialogue understanding on GPT-based foundational model.

**Weaknesses:**

The paper and background is slightly bias oriented to understanding, although in order to move the SOTA in dialogue modelling in human computer interaction, the faithful generation of spoken response is also critical. The paper lists the attributes pursued in the synthetic generation, and they dedicated effort in choosing sufficient good tools to generate the response, but there is not formal assessment of how faithful and suitable those target realizations are realistic. Without a human preference assessment between actual dialogues and the synthetic generated ones, the data set and benchmark presented in this work can limit the ceiling truth of the models developed using it as a benchmark. Still the work is valuable and will contribute to accelerate the foundational properties a the pre-training stage of Foundation models that power spoken dialogue systems.

The authors based there decisions on the audio generation side based only superficial knowledge of speech. Even their statements about the speech bandwidth is inaccurate and not scientifically supported. Authors should expand, improved and detailed describe the process and decision making on the fidelity and other speech attributes.

**Questions:**

a) How do we ensure the synthetic data is faithful respect to human-human interactions? the emotion modelling and its realization is not a 1-1 mapping and its suitability given the input emotion from the user of the system is not trivial.

b) Speech and spoken language communication is a 1 to many problem. Authors seem to define 1 single realization of a dialogue as the "ground-truth". Although multiple performances of the same dialogue (from the "speech generation" perspective would be acceptable. How authors and VoxDialogue datasets enable to consider the stochastic nature of human communication?

c) How VoxDialogue is scalable to mutliple languages and low resource languages?

---

> ### Author Response · Authors · 2024-11-22
> **Response to Reviewer NcGN**
>
> Thank you for your recognition of our work and for acknowledging the importance of our contributions to understanding paralinguistic information. Allow me to clarify some of your questions:
>
> **Q1**: Human Preference Evaluation
>
> **A1**: Thank you for your suggestion. We have manually filtered the synthesized data to ensure it aligns with human conversational standards as much as possible. However, collecting real spoken dialogue data is particularly challenging, especially when constructing and simulating scenarios that emphasize various paralinguistic information, as highlighted in this paper. This scarcity of real dialogue data has made it difficult for us to find sufficient real dialogue data for conducting human preference comparison on all attributes.
>
> Based on your advice, we conducted some evaluations on emotional data. Through an A/B test with DailyTalk[1] data, our data received 47% human approval, demonstrating that the authenticity of our data is reasonably assured to some extent.
>
> **Q2**: Fidelity Processing
>
> **A2**: Thank you for your suggestion. In the latest version, we have revised the description of the construction process for simulating poor audio fidelity:
>
> > For fidelity, according to the Nyquist-Shannon sampling theorem, the sampling rate must be at least twice the highest frequency of the signal to ensure lossless reconstruction. To capture frequencies up to 4 kHz, the minimum sampling rate should be 8 kHz. Therefore, we downsampled the speech to 4 kHz (to simulate the loss of speech signal and represent ‘poor’ audio quality, resulting in the loss of some speech information) and then resampled it back to 16 kHz to simulate poor audio fidelity.
> >
>
> Thank you for your valuable suggestions on our work. We have indeed put a lot of thought into constructing processes for each attribute to ensure the data reflects the corresponding acoustic properties. If you have any further writing suggestions or find any ambiguous points in our explanations, we look forward to further discussion.
>
> **Q3**: Spoken Dialogue is a One-to-Many Problem—How to Evaluate Reasonably?
>
> **A3**: Yes, spoken dialogue is a highly open-ended process where multiple responses can be considered correct. To make our evaluation system more rigorous, we referred to evaluation approaches from previous works [2,3] and chose to evaluate only the final response in multi-turn dialogue data. Under this constraint, the response must align with the preceding context, thereby limiting the openness of possible answers and enabling more effective evaluation.
>
> **Q4**: How can the dataset be extended to include more languages?
>
> **A4**: As long as corresponding TTS models are available, language-specific evaluation benchmarks can be generated. However, due to the significant variability in how different models perform with other languages, adding additional languages would require designing separate evaluation metrics for each language to effectively and fairly measure model performance on a per-language basis. This approach would ensure that each language is evaluated comprehensively and accurately.
>
> Thank you once again for your insightful discussion of our work. Your feedback has been invaluable, and we hope this response addresses your concerns. Should you have any further questions, we are always open to continued dialogue.
>
> [1] Li Y, Su H, Shen X, et al. Dailydialog: A manually labelled multi-turn dialogue dataset[J]. ACL 2017
>
> [2] Lin G T, Chiang C H, Lee H. Advancing large language models to capture varied speaking styles and respond properly in spoken conversations[J]. ACL2024
>
> [3] Ao J, Wang Y, Tian X, et al. SD-Eval: A Benchmark Dataset for Spoken Dialogue Understanding Beyond Words[J]. NeurIPS2024

---

### Official Review · Reviewer_KJyA · 2024-11-03

**Soundness:** 2
**Presentation:** 3
**Contribution:** 3
**Rating:** 3
**Confidence:** 4

**Summary:**

Voice assistants are systems that interact with human beings, but current systems usually only focus on linguistic information from text, neglecting useful verbal cues. This work provides a benchmark to evaluate current multimodal systems. In addition, they also identified 12 features that highly correlated to acoustic information and evaluated other dialogue systems on these features.

**Strengths:**

Other than semantic information from text, the authors proposed three directions worth investigating in dialogues: speaker information, paralinguistic information, and background sounds, which are valuable and well-designed.

**Weaknesses:**

1. On page 2, around line #100, 'For each of these attributes, we designed the most appropriate spoken dialogue synthesis pipelines' This needs more clarification; what is 'most appropriate'? How is it defined?
2. The second and third contributions listed at the end of the 'Introduction' section need a little more details, such as metrics for evaluating performance. From the description of the third contribution, it is unclear how and why the way to construct spoken dialogue data is unique/beneficial.
3. For the first two paragraphs under section 2.1, what are the differences between 'audio-language models' and 'spoken dialogue models'? There are no clear differences between the listed works and why they were separately discussed. In other words, I cannot find reasons to use two separate paragraphs, especially since they are all under the 'Spoken Dialogue System' subsection.
4. I think the third paragraph under section 2.1 is not appropriately placed. This subsection mainly discusses spoken dialogue systems, but not why the lack of a comprehensive benchmark for different evaluation tasks. It will be more appropriate to place it at end of section 2.2. Currently, the contents for the last paragraph from sections 2.1 and 2.2 are heavily overlapped.
5. I am confused for Table 2. For example, based on the examples given, 'business tasks' is dependent on 'man voice', and 'free juice' or/and 'beef burger' is dependent on 'young voice'. From the manuscript, I did not see how this is established for the speaker's information.
6. Many details are missing under section 3. Under section 3.2, authors state that they are referring to [Lin et al., 2024a] for their implementation of LLMs with advanced reasoning capabilities to synthesize spoken scripts. They did not specify the exact reason for this, especially what LLMs were used. It is not clear if GPT-4o is the only one that has been used or if it is one among a few. Also, under stage 2, 'We carefully tailored the most appropriate speech synthesis method for each attribute during the generation process,' what does 'most appropriate' even mean here? And how was it compared? For instance, if another work also considered paralinguistic information for their data synthesis process, why is your approach more 'appropriate' than theirs? Under stage 4, the authors state that 'For attributes such as volume, fidelity, audio events, and music, we performed post-processing to ensure that the audio aligns with the required expectations.' how do we interpret 'music' as an attribute here? In addition, 'For music, we randomly apply two different methods to integrate the music with the dialogues.' What does this even mean? How is it post-processed to align with the required expectations? How is music processed? What exactly is the expectation?
7. At the end of section 4.1 for task definition, is there a particular reason to use only the last response from the entire dialogue history for evaluation of the spoken dialogue systems?
8. It is not counter-intuitive that ASR-based systems perform poorly compared to multimodal systems because they only take text at input. The authors demonstrate (from the abstract section, conclusion section, and all the experiments) that ASR systems fail to capture important acoustic signals; it is never a fair comparison in the first place.

Minor issue:
The appearance of punctuation in the subsection titles could be more consistent. Why do sections 2.1 and 2.2 have periods in the titles?

Overall, many details and justifications are unclear and missing. The manuscript cannot be accepted in its current form.

**Questions:**

Please refer to the weaknesses listed.

---

> ### Author Response · Authors · 2024-11-22
> **Response to Reviewer KJyA (1/N)**
>
> Thank you for your recognition of the significance of the 12 key acoustic information aspects of our work and its overall value. Allow me to clarify your questions and address any concerns:
>
> **Q1**: What is ‘*most appropriate*’ spoken dialogue synthesis pipeline for each attribute? (weakness 1, 2, 6)
>
> **A1**: We noted your comment: “*If another work also considered paralinguistic information for their data synthesis process, why is your approach more ‘appropriate’ than theirs?*” We believe there may have been some misunderstanding. By ‘*most appropriate*,’ we mean that different attributes require distinct synthesis methods tailored specifically for their needs. This does not imply that our method is inherently more suitable than those proposed in other works.
>
> That said, we have indeed introduced new pipelines for certain acoustic attributes that have not been previously explored. These novel methods help address existing gaps in the field and offer fresh perspectives on the synthesis and control of various acoustic features in spoken dialogue. In the case of age, for instance, there is currently no TTS model capable of directly controlling vocal age. To address this, we use reference voices of different ages and apply zero-shot TTS models to achieve age-specific control. As detailed in Section 3.2, Stages 2 and 4 of our work, we outline the synthesis requirements for different attributes and propose five distinct synthesis methods using various TTS models to meet these needs effectively.
>
> Previous works [1,2] have made similar attempts but with notable limitations. For instance, SD-eval [1] used zero-shot TTS to generate adult voice data but did not achieve synthesis for children or elderly voices. StyleTalk [2] focused on emotional spoken dialogue synthesis but left many other attributes unexplored. To the best of our knowledge, our work is the first to comprehensively explore 12 different acoustic attributes in spoken dialogue, including several attributes (e.g., language, volume, speed, fidelity, stress, and non-verbal expressions) that have not been systematically studied in spoken dialogue tasks. As noted by other reviewers:
>
> > It is creative to use GenAI to create spoken data for target dimensions related to Speech.
> >
>
> We hope that our exploration of these diverse attributes will enhance the ability of spoken dialogue models to comprehensively understand and generate spoken dialogue across multiple dimensions.
>
> **Q2**: Difference between audio-language models and spoken dialogue models (weakness 3)
>
> **A2**: As described in Qwen-Audio [3]:
>
> > “Qwen-Audio, a fundamental multi-task audio-language model that supports various tasks, languages, and audio types, serving as a universal audio understanding model.”
> >
>
> > “Building upon Qwen-Audio, we develop Qwen-Audio-Chat through instruction fine-tuning, enabling multi-turn dialogues and supporting diverse audio-oriented scenarios.”
> >
>
> We consider an audio-language model to be a foundational model primarily focused on audio understanding, whereas spoken dialogue models build upon audio-language models and are specifically designed for multi-turn dialogue interactions. We apologize for any confusion this may have caused. Due to the frequent overlap between these models as different versions within the same project, we have included clarifications in the latest version to clearly distinguish these model types and their respective purposes.
>
> **Q3**: Explanation of examples (weakness 5)
>
> **A3**: When using ChatGPT-4o to generate spoken dialogue data, it was observed that the model, drawing on common market research, made the following recommendations based on shopping habits across different genders and ages:
>
> - Men typically prioritize features such as suitability for business travel and long battery life when purchasing smartphones.
> - Women often pay more attention to the aesthetics, tactile feel, and camera performance of smartphones.
> - Young people tend to prefer foods with stronger flavors, such as fruit juices and beef burgers.
> - Elderly individuals are more inclined to choose foods that are easy to digest and nutritious.
>
> These recommendations are based solely on general trends from market research and statistical data, aimed at generating suggestions without bias. We believe that detailed user information could further rationalize these recommendations; however, this goes beyond the primary focus of our work, which centers on age, gender, and other acoustic information. We have conducted manual checks to ensure the data does not contain strong group discrimination.

---

> ### Author Response · Authors · 2024-11-22
> **Response to Reviewer KJyA (2/N, N=2)**
>
> **Q4**: Why are audio and music considered attributes? (weakness 6)
>
> **A4**: We refer to audio and music as attributes because they are forms of acoustic information that contribute to spoken dialogue beyond words. Although not properties of the spoken language itself, they can influence spoken dialogue as environment information. We apologize if this terminology was misleading, and we will revise it to describe these as environmental attributes. We welcome your suggestions on further improvements.
>
> **Q5**: What does ‘meeting the desired expectations’ mean? (weakness 6)
>
> **A5**: As shown in Table 3, each speech instance in the spoken dialogue dataset has corresponding attribute labels. For attributes such as volume, fidelity, audio events, and music, it is not possible to directly use TTS models to generate audio that perfectly matches the labels. Therefore, additional post-processing is required to meet these expectations. For example, for samples labeled as “Low Fidelity,” we must transform normal fidelity audio into low fidelity through specific processing methods.
>
> Additionally, we have clarified the processing of music in the latest version as follows:
>
> > “For music, we randomly spliced it before the first speech segment or set it to play in a loop as background sound.”
> >
>
> **Q6**: Why only evaluate the final response? (weakness 7)
>
> **A6**: Understanding context is crucial for spoken dialogue systems. A short context sequence may simplify the task into a basic QA exercise, which does not require substantial contextual understanding. We consistently use the final response for evaluation to highlight the model’s capability to comprehend and utilize context effectively.
>
> Additionally, spoken dialogue is a highly open-ended process where multiple responses can be considered correct. To make our evaluation system more rigorous, we referred to evaluation approaches from previous works [1,2] and chose to evaluate only the final response in multi-turn dialogue data. Under this constraint, the response must align with the preceding context, thereby limiting the openness of possible answers and enabling more effective evaluation.
>
> **Q7**: ASR-based vs. spoken dialogue models (weakness 8)
>
> **A7**: In our experiments, it is important to note that spoken dialogue models did not outperform ASR-based systems across all metrics. ASR-based systems demonstrated robust context understanding capabilities, surpassing direct spoken dialogue models in certain attributes. Although ASR-based models may have limitations in understanding acoustic information, comparing them provides a valuable performance reference, representing the upper bound performance without the integration of acoustic information.
>
> **Q8**: Writing Issues (weakness 3,4)
>
> **A8**:  We have made the following improvements based on your suggestions:
> - We have revised the related works section to emphasize the importance of acoustic information in spoken dialogue models, especially at the end of Section 2.1.
>
> - Additionally, we have included more detailed descriptions of synthesized dialogue scripts and updated the explanation for the processing of music.
>
> - Lastly, we have removed the periods from the section titles in 2.1 and 2.2 to ensure consistency in formatting.
>
> Thank you for your attention and suggestions; they have enhanced the quality of our paper. We hope the updated version meets your expectations. If you have any additional feedback or questions, please feel free to reach out.
>
> [1] Ao J, Wang Y, Tian X, et al. SD-Eval: A Benchmark Dataset for Spoken Dialogue Understanding Beyond Words[J]. NuerIPS2024
>
> [2] Lin G T, Chiang C H, Lee H. Advancing large language models to capture varied speaking styles and respond properly in spoken conversations[J]. ACL2024
>
> [3] Chu Y, Xu J, Zhou X, et al. Qwen-audio: Advancing universal audio understanding via unified large-scale audio-language models[J]. arXiv2023

---

> > ### Author Response · Authors · 2024-11-26
> > **Looking Forward to Feedback Before the Paper Revision Deadline**
> >
> > We would like to express our sincere gratitude for your thorough reading and careful review of our manuscript. We hope that our responses adequately address all your concerns and resolve any misunderstandings. **As the deadline for submitting the revised paper is approaching, we eagerly await your feedback on the current version of our paper.**
> >
> > In response to your suggestions, we have made the following revisions:
> >
> > 1. **Revision to Contributions Section:**
> >
> >     In line with your suggestion #2, we have revised the wording of the last two points in the Contributions section (lines 120-129). By including performance metrics and additional details, we have clarified our contributions more effectively.
> >
> > 2. **Standardization of Terminology:**
> > Based on your suggestions #3 and #4, we have standardized the expression of “Spoken Dialogue System” in Section 2.1 (lines 142-154). Additionally, we have separated the introduction of the System and the Benchmark into Sections 2.1 and 2.2 to avoid any overlap in content.
> > 3. **Enhancements to Stage 4 Description:**
> > In response to your suggestion #6, we have supplemented the description of post-processing for each acoustic attribute in Stage 4 of Section 3 (lines 300-311) to enhance the scientific rigor and reproducibility of our work.
> > 4. **Polishing and Consistency Improvements:**
> >     - We have specified the version of the model in line 485.
> >     - We have improved the consistency of punctuation in subsection titles.
> >     - Based on feedback from other reviewers, we have further polished the overall writing to enhance the readability of the manuscript.
> >
> > We deeply appreciate your valuable suggestions for improving the writing of our paper. Regarding the technical issues you raised, we have provided detailed responses, and we hope that these replies have addressed all your concerns. If you have any additional questions or suggestions before the submission deadline, we will do our best to incorporate them.
> >
> > We look forward to receiving your feedback on the current version and are more than willing to make further improvements to our work.

---

> > > ### Comment · Reviewer_KJyA · 2024-11-26
> > > **Response to authors**
> > >
> > > Dear authors,
> > >
> > > Thank you for addressing the concerns I raised. Below are my follow-up responses. Please note that these responses correspond to the order of my original concerns, as I found it **difficult to follow** the authors' replies. Please ensure that the reply aligns with the order in which my concerns were raised (i.e., do not break weakness #6 into separate QA pairs).
> > >
> > > **1.** To my understanding, there are no content changes around line 100, as I see no highlighted edits made. While the authors do express the novelty of the dialogue systems in later sections, I find it unclear what is meant by "the most appropriate spoken dialogue synthesis pipelines" as I navigate toward the end of the introduction section. I understand that I will gain more insights into the proposed system later in the manuscript, but it is challenging to make connections at this moment. If the message cannot be clearly conveyed or connected, it might be better to exclude it to avoid confusing readers.
> > >
> > > **2.** Thank you for the changes made to the contributions listed. Although I cannot see or recall precisely how the original manuscript was written out, it is now clearer.
> > >
> > > **3.** Thank you for the clarifications; they are now clearer.
> > >
> > > **4.** Not addressed.
> > >
> > > **5.** Not addressed (I see some responses made under Q3, but please reorganize for clarity).
> > >
> > > **6.** The replies are hard to follow; please reorganize.
> > >
> > > **7.** Replied, but please reorganize.
> > >
> > > **8.** Is this reflected anywhere in the updated manuscript?
> > >
> > > Minor: Thank you for the modifications to the writing.

---

> > > > ### Author Response · Authors · 2024-11-27
> > > > **Futher Response to KJyA (1/N)**
> > > >
> > > > Thank you for your response and for recognizing the improvements we have made. We have further revised our paper, and please allow me to address your questions one by one:
> > > >
> > > > **R1:** In the latest version, we revised the description at line 100 to:
> > > >
> > > > > We designed a tailored spoken dialogue synthesis pipeline for each attribute to ensure that the synthesized dialogue data aligns accurately with the corresponding attribute.
> > > > >
> > > >
> > > > We hope this revised version eliminates any misunderstandings and look forward to your further suggestions.
> > > >
> > > > **R4:** Thank you very much for your valuable feedback. In the latest version, we have made further revisions to Sections 2.1 and 2.2 of the Related Work.
> > > >
> > > > - In Section 2.1, we provide an overview of existing spoken dialogue systems. In the final paragraph, we analyze the impact of acoustic attributes on dialogue and the limitations of current systems.
> > > > - In Section 2.2, we first highlight the contributions of existing benchmark studies. In the final paragraph, we summarize their shortcomings and emphasize the significance of our work in this area.
> > > >
> > > > We hope these revisions improve the overall structure of the paper. As the revision deadline is approaching, we will continue refining the manuscript based on your suggestions and strengthening it for the camera-ready version. We look forward to your further valuable feedback
> > > >
> > > > **R5:** When generating synthesized dialogue data, the model provides specific responses or recommendations based on different attributes (e.g., gender, age). To help clarify, we offer paired examples of reasoning behind these responses:
> > > >
> > > > - Gender
> > > >     - Men typically prioritize features such as suitability for business travel and long battery life when purchasing smartphones.
> > > >     - Women often pay more attention to the aesthetics, tactile feel, and camera performance of smartphones.
> > > > - Age
> > > >     - Young people tend to prefer foods with stronger flavors, such as fruit juices and beef burgers.
> > > >     - Elderly individuals are more inclined to choose foods that are easy to digest and nutritious.
> > > >
> > > > **R6-1: On referencing Lin et al., 2024a [1] to implement LLMs with advanced reasoning for synthesizing spoken dialogue scripts:**
> > > >
> > > > This work (Lin et al., 2024a [1]) was one of the first to explore using synthetic data to enhance a spoken dialogue system’s capability for emotional dialogue. Its data synthesis pipeline provides valuable insights. For our synthesis, we exclusively used GPT-4 to generate diverse spoken dialogue scripts.
> > > >
> > > > **R6-2: On the issue of ‘most appropriate’:**
> > > >
> > > > As discussed in **R1**, we will synchronize the changes in this description across all relevant sections. Thank you again for your valuable feedback. We hope these revisions address your concerns and look forward to your further input!
> > > >
> > > > **R6-3: “Why are audio and music considered attributes?”**
> > > >
> > > > We refer to audio and music as attributes because they are forms of acoustic information that contribute to spoken dialogue beyond words. Although not properties of the spoken language itself, they can influence spoken dialogue as environment information. We apologize if this terminology was misleading, and we will revise it to describe these as environmental attributes. We welcome your suggestions on further improvements.
> > > >
> > > > **R6-4: What does ‘meeting the desired expectations’ mean?  How is music processed? What exactly is the expectation?**
> > > >
> > > > As shown in Table 3, each speech instance in the spoken dialogue dataset has corresponding attribute labels. For attributes such as volume, fidelity, audio events, and music, it is not possible to directly use TTS models to generate audio that perfectly matches the labels. Therefore, additional post-processing is required to meet these expectations. For example, for samples labeled as “Low Fidelity,” we must transform normal fidelity audio into low fidelity through specific processing methods.
> > > >
> > > > Additionally, we have clarified the processing of music in the latest version as follows:
> > > >
> > > > > “For music, we randomly spliced it before the first speech segment or set it to play in a loop as background sound.”
> > > > >

---

> > > > ### Author Response · Authors · 2024-11-27
> > > > **Futher Response to KJyA (2/N, N=2)**
> > > >
> > > > **R7:** As emphasized in Lines 447-450, understanding context is crucial for spoken dialogue systems, which is reflected in our evaluation metrics. From an accuracy perspective, single-turn responses only test QA capabilities, while responses based on dialogue history require both multimodal understanding and contextual comprehension. From a stylistic perspective, maintaining a consistent speaking style throughout the conversation is also a significant challenge. As the dialogue progresses, an effective model should ensure stylistic consistency, reduce unfamiliarity, and foster a natural conversational flow.
> > > >
> > > > Furthermore, spoken dialogue is inherently open-ended, with multiple valid responses possible. To make the evaluation more rigorous, we followed established approaches from previous works [1,2] and chose to evaluate only the final response in multi-turn dialogue data. This approach ensures that the response aligns with the preceding context, reducing the openness of potential answers and enabling a more consistent and effective evaluation.
> > > >
> > > > **R8**: Thanks for your valuable feedback. We have added a discussion addressing this issue in the second part of Section 4.3. We believe that ASR-based models provides a valuable performance reference, representing the upper bound performance without the integration of acoustic information. Therefore, we suppose that comparing with such systems is meaningful.
> > > >
> > > > [1] Lin G T, Chiang C H, Lee H. Advancing large language models to capture varied speaking styles and respond properly in spoken conversations[J]. ACL2024
> > > >
> > > > [2] Ao J, Wang Y, Tian X, et al. SD-Eval: A Benchmark Dataset for Spoken Dialogue Understanding Beyond Words[J]. NuerIPS2024

---

> > > > ### Author Response · Authors · 2024-12-01
> > > > **Looking Forward to Your Further Feedback**
> > > >
> > > > Dear Reviewer KJyA,
> > > >
> > > > We sincerely thank you for your valuable suggestions on our work. Based on your insightful feedback, we have revised the manuscript and provided additional clarifications for descriptions that might have caused misunderstandings. We hope our responses have fully addressed your concerns.
> > > >
> > > > As the rebuttal deadline approaches, we kindly look forward to receiving your further feedback. Once again, we deeply appreciate the time and effort you have dedicated to reviewing our paper.
> > > >
> > > > Best regards,

---

> > > > > ### Comment · Reviewer_KJyA · 2024-12-03
> > > > > **Response to authors**
> > > > >
> > > > > Dear authors,
> > > > >
> > > > > Thank you for the revisions made. I have increased my scores for presentation and contribution, and I will consider updating the overall rating after a final detailed review.

---

### Official Review · Reviewer_mXjg · 2024-11-04

**Soundness:** 4
**Presentation:** 4
**Contribution:** 3
**Rating:** 8
**Confidence:** 5

**Summary:**

This paper provides a dataset involving a spoken dialogue corpus between two humans. It is designed to evaluate spoken dialogue systems. The data is selected so as to include 12 different characteristics where speech would help, such as emotion.

**Strengths:**

Overall, this is a clean paper proposing a valuable dataset for spoken language researchers. It is creative to use GenAI to create spoken data for target dimensions related to Speech. I also opened the examples in the github repo and it is very possible that this dataset will be employed by many researchers in this field.

**Weaknesses:**

The dataset does not include any task oriented dialogue. Hence the evaluation is limited to BLEU or GPT ratings. In many real life scenarios, the spoken dialogue systems are aiming at either an agent like scenario, like Google Home/Alexa style personal assistants, or call center automations, or outbound calls. Their performance cannot be evaluated using BLEU only and the target goal completion is critical. Maybe in the next version of the dataset the authors may want to extend this dataset with such data.

**Questions:**

- the dataset may also be expanded with a target goal as in MultiWoz / ATIS. For example there is a frustrated call center example but the content is not there, it is just emotional exchanges.
- can you elaborate further and think of some ablations especially regarding the difference of FunAudioLLM from direct dialogue models? why exactly is this model performing better and in which dimensions?

---

> ### Author Response · Authors · 2024-11-22
> **Response to Reviewer mXjg**
>
> Thank you for your positive score and thoughtful insights on our work, recognizing the value of our dataset. Allow me to address the questions you raised and discuss our considerations.
>
> **Q1:** Discussion on Task-Oriented Dialogue
>
> **A1:** We fully agree with your perspective on the significance of task-oriented dialogue. Initially, we had ever considered constructing our dataset based on task-oriented dialogues (such as MultiWoz [1] and SpokenWoz [2]). However, after careful deliberation, we decided to put this in the next stage of our research. The current spoken dialogue tasks can be categorized into daily dialogues (e.g., Daily Dialogue [3], MELD [4], focusing on emotion and content comprehension) and task-oriented dialogues (e.g., MultiWoz, SpokenWoz, emphasizing task objectives and dialogue state tracking). However, we realized that these task-oriented dialogues often prioritize content information (the main consultation content) and contextual memory (user-specific information), rather than the acoustic features emphasized in this paper. As a result, we opted for daily dialogue scenarios, which are more sensitive to acoustic nuances.
>
> However, your insightful example has underscored the importance of acoustic information in real-world scenarios. Following your advice, we will explore whether spoken dialogue models can more effectively handle task-oriented dialogues while leveraging acoustic features in our future work. Thank you for your valuable suggestion.
>
> **Q2:** FunAudioLLM vs. Spoken Dialogue Models
>
> **A2:** FunAudioLLM processes speech by converting it into text using ASR, which is then passed into a large language model (LLM) for inference. During training, LLMs are likely exposed to extensive textual dialogue data, enhancing their ability to infer attributes such as age, gender, timbre, and language based on context on dialogue tasks. In contrast, as noted by Tang et al. [5], task overfitting remains a significant challenge in multi-modal LLM training. Current audio-language models primarily focus on audio comprehension and often lack training data formatted for spoken dialogues, which hampers their performance in dialogue-specific tasks.
>
> This discrepancy explains why spoken dialogue models currently underperform compared to FunAudioLLM in dialogue tasks. A viable solution would be to leverage synthetic data to strengthen the dialogue capabilities of these models. Our proposed spoken dialogue synthesis pipeline is well-positioned to support such efforts by providing diverse and attribute-specific synthetic dialogue data.
>
> Thank you once again for your insightful discussion of our work. Your feedback has been invaluable, and we hope this response addresses your concerns. Should you have any further questions, we are always open to continued dialogue.
>
> [1] Si S, Ma W, Gao H, et al. Spokenwoz: A large-scale speech-text benchmark for spoken task-oriented dialogue agents[J]. NeurIPS 2023
>
> [2] Budzianowski P, Wen T H, Tseng B H, et al. Multiwoz--a large-scale multi-domain wizard-of-oz dataset for task-oriented dialogue modelling[J]. ACL 2018
>
> [3] Li Y, Su H, Shen X, et al. Dailydialog: A manually labelled multi-turn dialogue dataset[J]. ACL 2017
>
> [4] Poria S, Hazarika D, Majumder N, et al. Meld: A multimodal multi-party dataset for emotion recognition in conversations[J]. ACL 2019
>
> [5] Tang C, Yu W, Sun G, et al. Salmonn: Towards generic hearing abilities for large language models. ICLR 2024

---

> > ### Comment · Reviewer_mXjg · 2024-11-26
> >
> > Thanks, I am keeping my score

---

### Official Review · Reviewer_mqdD · 2024-11-04

**Soundness:** 3
**Presentation:** 3
**Contribution:** 3
**Rating:** 8
**Confidence:** 3

**Summary:**

This paper introduces a new benchmark called VoxDialogue for evaluating the audio comprehension capabilities of voice dialogue systems. The benchmark encompasses 12 sound-related attributes, including speaker attributes (age, gender, accent, language), paralinguistic features (emotion, volume, speed, fidelity, stress, non-verbal expressions), and background sounds (audio, music). The paper also conducted a systematic evaluation of existing spoken dialogue systems, comparing their performance in terms of understanding acoustic information. Besides, the paper proposed a comprehensive method for constructing spoken dialogue data tailored to different acoustic attributes, enabling large-scale data synthesis to support model training.

**Strengths:**

1. This paper addresses the current issue of speech dialogue systems ignoring audio information by proposing a new benchmark test, which is a meaningful research endeavor.

2. The evaluation of ASR-Based/Direct Spoken Dialogue Systems reveals the limitations of current ASR-based and direct speech dialogue models.

3. The construction of a challenging test set containing 4.5K multi-turn dialogue samples can provide assistance to the voice dialogue systems research community.

**Weaknesses:**

1. Recommend to add the specific values of text generation metrics in the appendix.

2. Suggest to include statistics on the duration of the dataset.

**Questions:**

What do AVG and Dur. refer to in Table 3?

---

> ### Author Response · Authors · 2024-11-22
> **Response to Reviewer mqdD**
>
> Thank you for your high evaluation of our work and for recognizing the value of the 12 acoustic attributes in spoken dialogue systems, as well as the significance of our proposed methods for constructing spoken dialogue data tailored to these attributes. Allow me to share the updates we made based on your review:
>
> - In Table 5, we have added the specific values for each metric in Figure 2 to provide clearer references.
> - In Table 3, avg represents the average number of dialogue turns, and dur indicates the total duration of the audio files (in hours). We have clarified these definitions in the table caption for better understanding.
> - Following your suggestion, we included the distribution of dialogue durations in Figure 1c, visually illustrating the length distribution of dialogue.
>
> Thank you again for your valuable feedback and for recognizing the contributions of our work. Your suggestions have helped us refine and improve our paper further. We hope these updates address your concerns. If you have any additional questions or suggestions, please feel free to reach out to us.

---

> > ### Comment · Reviewer_mqdD · 2024-11-28
> >
> > Thank you for your reply. I maintain my score.

---

### Author Response · Authors · 2024-12-04
**General Response**

Thank you for your invaluable efforts and constructive feedback on our work. As the rebuttal period comes to a close, we would like to take this opportunity to summarize the discussions and outline the improvements we have made based on your suggestions.

## **Major Contributions of Our Work**

We are grateful for the reviewers’ recognition of our work. Below, we summarize the key contributions highlighted:

1. **Motivation:**
    - This work introduces a novel spoken dialogue system benchmark, VoxDialogue, to explore whether current systems can comprehend various acoustic properties. To the best of our knowledge, we are the first to provide such a comprehensive benchmark, making it a valuable resource to advance research in spoken dialogue systems. **[mqdD, mXjg, KJyA, NcGN, CNoA]**
    - We systematically analyzed 12 acoustic attributes that may influence spoken dialogue responses. For each of the 12 acoustic attributes, we designed a dedicated data synthesis pipeline with careful verification to ensure controllable manipulation of these attributes. **[mqdD, mXjg, NcGN, CNoA]**
2. **Experiments:**
    - All reviewers agreed that our benchmark represents a fair and well-designed experiment, providing an objective evaluation framework for spoken dialogue systems. **[mqdD, mXjg, KJyA, NcGN, CNoA]**
    - The experiments conducted on five distinct spoken dialogue systems are highly insightful, highlighting the limitations of both ASR-based systems and direct spoken dialogue systems. **[mqdD, CNoA]**

3.	**Presentation and Impact:**
    - All reviewers commended the clarity and quality of our demonstration. **[mqdD, mXjg, KJyA, NcGN, CNoA]**
    - This work lays a solid foundation for the standardization, democratization, and acceleration of spoken dialogue system evaluations. It provides a thorough review of the current state-of-the-art while comprehensively addressing speech attributes and dialogue rules. **[mqdD,mXjg,NcGN,CNoA]**

## **Revisions and Improvements Based on Reviewer Suggestions**

In response to the reviewers’ insightful comments, we have revised and enhanced the manuscript with the following updates:

1. **Additional Explanations**
    - We provided more detailed explanations for potentially confusing attributes in the benchmark, emphasizing that music is an environmental information beyond the language itself, and refining the construction details for fidelity. **[NcGN,KJyA]**
    - We elaborated on the design rationale behind the synthetic scripts and GPT evaluation metrics. Additionally, we included all relevant prompts in the appendix to clarify the details of our LLM-based methodology. **[CNoA,KJyA]**
2. **Improved Figures and Tables**
    - We added Figure 1(c) to illustrate the distribution of dialogue duration, further clarifying the details of our benchmark. In addition, we provided Table 5 in the appendix to supplement Figure 2 with detailed evaluation metrics, enhancing the credibility of our experimental results. **[mqdD]**
    - We added a comparison with LeBenchmark in Table 1. **[CNoA]**

3. **Writing Refinements**

    - We clarified that the baseline experiments used the Chat version and further emphasized the importance of acoustic information in spoken dialogue models. We also unified the punctuation throughout the titles of each subsection. **[KJyA]**
    - We defined and explained abbreviations such as AVG and Dur., improving the readability of the paper. **[mqdD]**

4. **Fairness and Ethical Discussions**

    - We reviewed the literature on “dialogue bias” and revised potentially offensive expressions in the VoxDialogue dataset. **[CNoA]**
    - Additionally, we added an Ethical Discussion section in Appendix C to further address fairness-related concerns, enhancing the usability and societal impact of our work. **[CNoA]**

## **Conclusion**

Once again, we deeply appreciate the reviewers’ dedication and valuable feedback, which have greatly contributed to refining this work. We are honored by the opportunity to present our research and look forward to any further suggestions.

---

### Meta-Review · Area_Chair_5S8M · 2024-12-22

**Metareview:**

This paper proposes a benchmark, VoxDialogue, for evaluating the ability of spoken dialogue systems to understand acoustic information beyond lexical content. VoxDialogue integrates12 acoustic dimensions, including speaker attributes (age, gender, accent, language), paralinguistic features (emotion, volume, speed, fidelity, stress, non-verbal expressions), and environmental information (audio, music). The paper includes comparison of VoxDialogue to several other benchmarks with respect to many evaluation dimensions. The work also proposes spoken dialogue data synthesis methods for various acoustic attributes and compares the performance of existing spoken dialogue systems in terms of understanding acoustic information using a GPT-based metric. Reviewers acknowledged that these contributions of the paper constitute its strengths. They also asked several clarification questions and requested explanations which the authors responded to during the rebuttal period, and integrated them into the paper already.

**Additional Comments On Reviewer Discussion:**

One of the reviewers mentioned they will increase their score, but it stayed as 3. I took a higher overall rating from the specific reviewer as a bases in my assessment of the paper.

---

### Decision · Program_Chairs · 2025-01-22

Accept (Poster)